# Status and related factors of medication safety behaviour of nurses in the operating room: A cross-sectional survey in China

Xiuwen Chen[1,2,3], Xueyi Wei[1], Liqing Yue[2,3]*, Duo Wu[4], Jiqun He[2]

**1** Xiangya School of Nursing, Central South University, Changsha, China, **2** Teaching and Research Section of Clinical Nursing, Xiangya Hospital, Central South University, Changsha, China, **3** National Clinical Research Center for Geriatric Disorders, Xiangya Hospital, Central South University, Changsha, China, **4** Department of Nursing, The Xiangya Hospital, Central South University, Changsha, China

These authors contributed equally to this work.
* ylq6998@163.com

## Abstract

### Aim

This study aimed to investigate the current status and influencing factors of medication safety behaviour of operating room nurses.

### Background

Medication safety is an important safety issue recognised by health organisations all over the world, and the operating room has one of the highest rates of preventable medication-related injuries. However, few studies have investigated the current status and influencing factors of medication safety behaviour of operating room nurses.

### Design

A cross-sectional study.

### Methods

From February to March, this study was conducted at three tertiary hospitals in southern China. Operating room nurses completed a series of questionnaires, including nurses' medication environment perception scale and operating room nurse's medication safety behaviour questionnaire. Data analysis included descriptive statistics, ANOVA, correlation analysis, and multiple regression. The STROBE checklist guided the reporting of this study.

### Results

A total of 171 questionnaires were analysed. The total score on medication safety behaviour of operating room nurses was 78.20±8.94. The medication environment

**Data availability statement:** All data generated or analysed during this study are included in this published article.

**Funding:** This work was supported by the 2024 Annual Graduate Student Independent Exploration and Innovation Project of Central South University (2024ZZTS0548). There is no role in study design, data analysis, decision to publish, or preparation of the manuscript.

**Competing interests:** The authors declare no conflict of interest.

the operating room nurses perceived was positively correlated with medication safety behaviour ($P < 0.01$). Additionally, factors related to the medication safety behaviour of operating room nurses included working years($B = 4.899$, $P = 0.000$), the highest level of education($B = 5.440$, $P = 0.000$), professional title($B = -2.644$, $P = 0.002$), the last time of nursing medication safety training($B = -0.914$, $P = 0.013$), and the *system and supervision* ($B = 0.141$, $P = 0.015$).

## Conclusion

The medication safety behaviour of operating room nurses is low. The relationship between individual factors, medication environment, and medication behaviour of operating room nurses should be deeply considered, and targeted intervention strategies should be carried out to influence factors to improve their medication safety behaviour.

## 1 Introduction

The World Health Organization (WHO) launched the global patient safety challenge on medication safety in 2017, which aimed to reduce serious and avoidable medication-related harm by 50% in all countries over the next five years by addressing the weaknesses in health systems that lead to medication errors and the severe harm that results [1]. Medication safety is an important safety issue recognised by health organisations worldwide [2]. Medication administration errors (MAEs) refer to drug administration errors caused by deviation from the doctor's order or prescription content in the case of correct prescription [3]. Studies have shown that the incidence of MAEs is about 10%, ranging from approximately 5% to 20% of medication administrations in hospitals, and MAEs are the leading cause of patient harm [4]. Costs associated with medication errors are estimated at $42 billion annually globally [5]. MAEs may occur at all stages of drug administration, and unsafe medication behaviour of clinical nurses is the main factor leading to MAEs [6,7]. Therefore, improving the ability of nurses to safely administer medication is crucial to achieving this initiative.

The operating room (OR) is the setting for the most invasive treatments in the hospital, where high-risk drugs such as narcotic, toxic, and powerful drugs are routinely administered, and the incorrect selection or improper use of high-risk drugs is likely to result in serious accidents [8]. Most surgical patients cannot express any discomfort following medication administration because they are sedated during the operation, making the consequences of medication misuse particularly severe. The Institute for Safe Medication Practices (ISMP) has reported serious medical incidents resulting in severe burns and even death due to intraoperative medication errors [9]. Most studies have found that intraoperative medication errors pose potential threats to patient safety and even endanger patients' lives, and failure to implement safety strategies will cause such risks to persist for a long time [10,11]. In addition, the high-pressure, fast-paced, and tense working environment in the operating room, coupled with frequent oral medical instructions, increases the likelihood

of medication errors [12]. A prospective observational study of medication errors in OR showed that 193 out of 3 671 MAEs occurred in OR, with an incidence of 5.3%, and 79.3% were preventable [13]. Additionally, studies have pointed out that one medication administration error occurs in every one operation or every 20 dosing, and nearly one-third of MAEs will cause harm to patients [12,14]. Despite this, the operating room remains one of the areas with the highest rate of preventable medication-related injuries [15]. This highlights the need to pay attention to medication safety in the operating room to reduce or eliminate medication errors.

Operating room nurses play a crucial role in drug administration, including dispensing, checking, and assisting the anesthesiologist in administering drugs [8]. They are not only the observers of the reaction after drug administration but also the implementors and protectors of safe drug use, shouldering important responsibilities in ensuring drug safety and also the last line of defence to eliminate medication errors in the operating room [16]. Their behaviour is directly related to the occurrence of MAEs. However, in recent years, studies on medication safety in OR have mainly focused on investigating incidence, monitoring medication errors, reporting methods for MAEs, and evaluating the effectiveness of quality management [17–19]. There have been few studies investigating the medication safety behaviour of OR nurses. Therefore, this study aimed to explore the status and related factors of medication safety behaviour of operating room nurses and to provide a basis for the formulation of medication risk management measures in OR.

## 2 Methods

### 2.1 Study design and participants

From February to March 2023, a cluster random sampling method was used to randomly select three tertiary hospitals in southern China. Specifically, the list of all three tertiary hospitals in southern China was compiled, and the hospitals were divided into several groups based on their geographical location and hospital scale. Next, the research team randomly selected three tertiary hospitals from the compiled list. Then, a cross-sectional study of operating room nurses from the selected three hospitals was conducted to investigate their medication safety behaviour. The Guidelines for the Strengthening the Reporting of Observational Studies in Epidemiology(STROBE) were used in this study [20].

The inclusion criteria are as follows: (1)having a nurse license; (2) having at least 1 year of clinical nursing experience; (3) working in the operating room; and (4) informed consent and voluntary participation in the study. Exclusion criteria: (1) nurse trainee; (2)unable to participate due to physical illness; and (3) nurses not on duty or vacation.

For the sample size, the recommendations are between 5 and 10 times the number of explanatory variables [21,22]. With an estimated 15 explanatory variables (10 demographic variables, five dimensions of nurses' medication environment perception scale) and accounting for 20% of invalid questionnaires, a minimum of 90 operating nurses was required.

### 2.2 Measurements

**2.2.1 Demographic information questionnaire.** According to the purpose of the research, the general information questionnaire was made, and the participants' demographic information included age, sex, the highest level of education, present position, professional title, working years, marital status, average monthly income, experience of medication safety training and the last time of nursing medication safety training.

**2.2.2 Nurses' medication environment perception scale.** The scale was compiled by Liu et al. [10]. based on the SHEL (Software, Hardware, Environment, Liveware) model to screen nurses' perception of the medication environment. The scale includes 32 items across five dimensions: *System and supervision* (11 items), *equipment* (4 items), *environment* (3 items), *nurses* (7 items), and *relevant personnel* (7 items). The *system and supervision* dimension refers to the perception of safety regarding rules, regulations, and management processes involved in the process of nurses' medication. The dimensions of *environment* and *equipment* refer to the safety perception of nurses' working environment and equipment used in the medication process, respectively. The dimension of *a nurse refers to the perception of safety regarding nurses' professional qualities*. The *relevant personnel* dimension refers to the safety perception of other

personnel (doctors, patients) related to the medication nurse. All items are scored on a five-point Likert-like scale, ranging from "strongly disagree" to "strongly agree", with a total score ranging from 32 to 160 points. The higher the score, the better the medication environment perceived. The Cronbach's alpha of the scale in this study was 0.979.

**2.2.3 Operating room nurses medication safety behavior questionnaire.** The questionnaire was designed by the research group based on the previous research and modified, discussed, and compiled by the group on the basis of extensive references to relevant literature. There are 28 items; the first two are true or false questions, "Yes" counts 1 point, and "No" counts 5 points. The last 26 questions are scored on a five-point Likert-like scale, with "never", "rarely", "sometimes", "often" and "always". The questionnaire is divided into five dimensions, including *medication placement* (4 items), *doctor's order and check* (8 items), *dispensing* (6 items), *aseptic principle*(3 items), *inspection and transport*(5 items), with a total score ranging from 28 to 140 points. The higher the score, the higher the medication safety behaviour of the operating room nurses. The Cronbach's alpha of the questionnaire was 0.678.

## 2.3 Data collection

Electronic questionnaires were used to collect data. The researchers first contacted the head nurses in the operating rooms of three randomly selected hospitals and briefly trained them on the methods of data collection and precautions. The questionnaire link was sent to 3 head nurses, and the head nurses sent the link to the nurses who met the inclusion and exclusion criteria. The questionnaire used unified guidance to explain the purpose of the survey, answer methods, filling requirements and precautions, etc. The results were directly returned to the researchers after the operating room nurses completed the questionnaire. To ensure the quality of the survey results, all researchers received uniform and pre-survey training, and researchers worked rigorously and communicated skillfully. Re-enter and check the data after collecting the questionnaire to ensure its authenticity and reliability.

## 2.4 Statistical analysis

Excel was used for data entry, and IBM SPSS v25.0 was used for data analysis. The measurement data were described by means and standard deviations, while the count data were expressed as frequencies and percentages. Statistical inference was carried out using the test level of α = 0.05. Bilateral probability values were taken as *P*-values. *T*-test and ANOVA were used for data with normal distribution and homogeneity of variance. Pearson correlation analysis was used for correlation analysis, and multiple linear regression was used for regression analysis. $p < 0.05$ was considered statistically significant. The scoring rate is the actual score (theoretical maximum(100%) [23]. The scoring rates greater than 80%, between 60% and 80%, and less than 60% indicate that the medication safety behaviour is good, medium and poor, respectively.

## 2.5 Ethical considerations

This study was approved by the Ethics Committee of Xiangya Hospital, Central South University (registry number 2022100085). The study complied with the Declaration of Helsinki; all participants signed informed consent forms and participated anonymously.

# 3 Results

## 3.1 Sociodemographic characteristics and univariate analysis of the study participants

A total of 174 questionnaires were distributed, while 171 questionnaires were valid, with an effective response rate of 98.3%. Among the participants, 27.5%, 22.2%, 34.5% and 15.8% worked for <6 years, 6–10 years, 11–20 years and >21 years, respectively. About 91.2% of the sample was female, and males accounted for 8.8%. 45.0% of nurses were aged 31–40, and 70.8% had bachelor's degrees. The main technical titles were supervisor nurses, accounting for 46.8%. About

86.0% of nurses had received medication safety training. The normality test showed that all the demographic variables in this study obeyed normal distribution except the highest level of education and average monthly income. Analysis of variance, Kruskal-Wallis test and *t*-test revealed that the demographic factors related to the medication safety behaviour of OR nurses were age, working years, highest level of education, average monthly income, present position, professional title, experience of medication safety training and the last time of nursing medication safety training ($p<0.05$). Other demographic details and univariate analysis results are shown in Table 1.

### 3.2 Medication environment perception of operating room nurses

This study showed that operating room nurses' total score of medication environment perception was 134.05±25.47, and the scoring rate was 83.8%. The scoring rate of each dimension from high to low was *system and supervision*, *nurses* and *environment, equipment*, and *related personnel.* Refer to Table 2 for the specific information of each dimension.

### 3.3 Medication safety behaviour of operating room nurses

The total score of medication safety behaviour of nurses in the OR was 78.20±8.94, and the scoring rate was 53.7%, which was moderate in *inspection and transport medication placement* but low in *doctor's order and check*, *dispensing* and *aseptic principle*. In addition, item 1 and item 2 showed that 35 participants (20.46%) have experienced medication administration errors, and 11 nurses (6.43%) have experienced medication administration errors that have caused disputes. The detailed information on each dimension score is shown in Table 3. Additionally, the three items with the lowest score were Q12. First aid drugs will be carried during the skin test, and drugs requiring a skin test will be administered after the skin test (1.28±0.746), Q17. Will not be based on experience to prepare drugs without understanding drug compatibility contraindications (1.32±0.717), Q18 Will not be to complete work tasks or save time in advance to prepare drugs without considering the drug time limit required by the instructions(1.35±0.821).

### 3.4 Correlation analysis between medication environment perception and medication safety behaviour of operating room nurses

Pearson correlation analysis showed that the total score of operating room nurses' perception of the medication environment was positively correlated with the total score of medication safety behaviour ($P<0.01$). The correlation between each dimension of operating room nurses' perception of the medication environment and each dimension of medication safety behaviour is shown in Table 4.

### 3.5 Multivariate analysis of medication safety behaviour of nurses in the operating room

The total score of medication safety behaviour of operating room nurses was taken as the dependent variable, the demographic variables were statistically significant in the univariate analysis, and the five dimensions of the medication environment perception scale were taken as the independent variables for multivariate analysis by stepwise regression method (Assignment of independent variables was shown in Table 5). The results showed that working years, the highest level of education, professional title, the last time of nursing medication safety training, *and system and supervision* were the related factors of medication safety behaviour of operating room nurses. See Table 5 for details.

## 4 Discussion

To the best of our knowledge, this study clarifies the current state and related factors of medication safety behaviour of OR nurses, thereby contributing valuable insights to the existing body of knowledge on this topic. Our study reveals that the overall medication safety behaviour of OR nurses was low, with the medication environment perceived by the OR nurses emerging as a significant influencing factor. Additionally, working years, the highest level of education, system and

**Table 1. Sample demographics and Univariate analysis (n = 171).**

| Variable | Category Group | Frequency (%) | Univariate analysis results |
|---|---|---|---|
| Age (years) | 21–30 | 57(33.3) | $F=8.027$ $P=0.000^*$ |
| | 31–40 | 77(45.0) | |
| | 41–50 | 28(16.4) | |
| | 51–60 | 9(5.3) | |
| Sex | Male | 15(8.8) | $t=-0.243$ $P=0.808$ |
| | Female | 156(91.2) | |
| Working years (years) | <6 | 47(27.5) | $F=13.983$ $P=0.000^*$ |
| | 6–10 | 38(22.2) | |
| | 11–20 | 59(34.5) | |
| | >21 | 27(15.8) | |
| Marital status | Married | 120 (70.2) | $F=0.284$ $P=0.753$ |
| | Spinsterhood | 45(26.3) | |
| | others | 6(3.5) | |
| Highest level of education | Associate degree | 43(25.1) | $H=14.379$ $P=0.001^*$ |
| | Bachelor degree | 121(70.8) | |
| | Master's degree and above | 7(4.1) | |
| Average monthly income(CNY) | <5000 | 32(18.7) | $H=30.814$ $P=0.000^*$ |
| | 5000–10000 | 98(57.3) | |
| | >10000 | 41(24.0) | |
| Present position | Nurse | 116(67.8) | $F=4.260$ $P=0.006^*$ |
| | Head nurse | 31(18.1) | |
| | Specialist group leader | 18(10.5) | |
| | Others | 6(3.5) | |
| Professional title | Nurse | 25(14.6) | $F=2.691$ $P=0.033^*$ |
| | Senior nurse | 38(22.2) | |
| | Supervisor nurse | 80(46.8) | |
| | Deputy chief nurse | 27 (15.8) | |
| | Chief nurse | 1(0.6) | |
| Experience in medication safety training | Yes | 147(86.0) | $t=-0.641$ $P=0.522$ |
| | No | 24(14.0) | |
| The last time nursing medication safety training | Within a month | 45(30.6) | $F=2.683$ $P=0.023^*$ |
| | Within three months | 41(27.9) | |
| | Within half a year | 27(18.4) | |
| | Within a year | 13(8.8) | |
| | More than a year | 21(14.3) | |

Note:

*indicates $P<0.05$; CNY stands for Chinese Yuan

supervision were positively correlated with the medication safety behaviour of OR nurses. Notably, their professional title and the last medication safety training were negatively correlated with their medication safety behaviour.

## 4.1 Medication safety behaviour of operating room nurses

The results of this study showed that the scoring rate of medication safety behaviour of OR nurses was at a low level, indicating that there are still certain limitations and deficiencies in the safe medication management process of OR nurses.

**Table 2. Scores of medication environment perception of operating room nurses(n = 171).**

| Dimension | Score (Mean ± SD) | Scoring rate(%) |
|---|---|---|
| System and supervision | 48.39±9.74 | 88.0 |
| Nurses | 30.78±6.02 | 87.9 |
| Environment | 13.18±2.69 | 87.9 |
| Equipment | 16.16±4.10 | 80.8 |
| Related personnel | 25.54±5.95 | 73.0 |

**Table 3. Scores of medication safety behaviour of operating room nurses(n = 171).**

| Dimension | Score (Mean ± SD) | Scoring rate(%) |
|---|---|---|
| Inspection and transport | 16.15±2.47 | 64.6 |
| Medication placement | 12.73±2.39 | 63.7 |
| Doctor's order and check | 22.35±3.15 | 55.9 |
| Dispensing | 12.51±3.27 | 41.7 |
| Aseptic principle | 5.53±2.38 | 36.9 |

**Table 4. Correlation analysis between medication environment perception and medication safety behaviour of operating room nurses (*r*-value).**

| Dimension | Medication placement | Doctor's order and check | Dispensing | Aseptic principle | Inspection and transport | Total score of medication safety behavior |
|---|---|---|---|---|---|---|
| System and Supervision | 0.270** | 0.150 | −0.083 | −0.161* | 0.468** | 0.218** |
| Equipment | 0.242** | 0.003 | −0.186* | −0.175* | 0.453** | 0.146 |
| Environment | 0.279** | 0.093 | −0.114 | −0.216** | 0.445** | 0.187* |
| Nurses | 0.270** | 0.125 | −0.122 | −0.174* | 0.484** | 0.197** |
| Relevant personnel | 0.189* | 0.213** | 0.052 | 0.005 | 0.317** | 0.248** |
| The total score of medication environment precipitation | 0.280** | 0.147 | −0.090 | −0.152* | 0.487** | 0.231** |

Note:

**indicates $P < 0.01$,

*indicates $P < 0.05$

To our knowledge, this is the first time such a study has been conducted, so it cannot be compared with previous findings. However, a previous study in Egypt that investigated the practice of general nurses in medication administration showed similar findings to this study [24]. In addition, A national survey of nurses in China showed a high proportion of unsafe medication behaviours among nurses [25]. Yu et al. [26] also showed that a considerable number of nurses experienced unsafe medication behaviours when administering drugs. However, the above research results are for reference only. The operating room environment's particularity makes the nurses' medication behaviour different from that of general nurses. In the future, a large-sample study should be carried out to investigate the medication behaviour of this specialised nurse. The OR is characterised by a large variety of high-risk drugs, high consumption, less management staff, less available storage space, and short drug preparation time etc, which makes the work of specialised nurses onerous [11]. Studies have shown that work pressure is one of the risk factors for nursing errors, and nurses in the operating room are under greater pressure than nurses in other departments [27].In addition, the intensive turnover of narcotic drugs and psycho-tropic drugs in the OR leads to a series of problems, such as difficult drug management and poor access to intraoper-ative drugs. According to the human error model developed by Professor Reason, unsafe medication behaviour is the

**Table 5. Assignment of independent variables and multivariate analysis of medication safety behaviour of nurses in the operating room (n = 171).**

| Variable | *B*-Value | Standard Error | *t* | *P* | Assignment |
|---|---|---|---|---|---|
| Working years | 4.899 | 0.753 | 6.502 | 0.000 | "<6" = 1; "6–10" = 2; "11–20" = 3; ">21" = 4 |
| Highest level of education | 5.440 | 1.118 | 4.864 | 0.000 | "Associate degree" = 1"; "Bachelor degree" = 2; "Master degree and above" = 3 |
| Professional title | −2.644 | 0.852 | −3.103 | 0.002 | "Nurse" = 1; "Senior nurse" = 2; "Supervisor nurse" = 3; "Deputy chief nurse" = 4; "Chief nurse" = 5 |
| The last time nursing medication safety training | −0.914 | 0.363 | −2.522 | 0.013 | "Within a month" = 1; "Within three months" = 2; "Within half a year" = 3; "Within a year" = 4; "More than a year" = 5 |
| System and supervision | 0.141 | 0.057 | 2.469 | 0.015 | "strongly disagree" = 1; "disagree" = 2; "neutrality" = 3; "agree" = 4; "strongly agree" = 5 |

direct cause of medication errors and may eventually lead to clinical adverse events [28]. Therefore, medication safety behaviour is one of the important indicators for managers to improve medication errors. Managers should strengthen operating room nurses' management of unsafe medication behaviour, especially the management of unsafe behaviours that will not lead to medication errors, which is crucial for controlling medication errors.

### 4.2 Factors related to medication safety for operating room nurses

#### 4.2.1 The medication environment perceived by the OR nurses.

The medication environment can indirectly affect nurses' medication safety behaviour by influencing their professional quality [29,30]. This study showed that the medication environment perceived by OR nurses positively correlated with medication safety behaviour, indicating that a good medication environment promotes nurses to perceive the hospital's safety culture and improve their medication safety ability. The nurse's work environment has long been recognised as an important and modifiable organisational feature that affects patient outcomes, encouraging nurses to think critically about medical treatment and sequences of care and make recommendations for care plans [31]. According to reciprocal determinism, the organisational environment is the basis of work behaviour and plays a guiding role in this behaviour [32]. The OR medication environment differs from that of an inpatient ward, and perioperative medication management often bypasses standard safety checks, such as electronic order entry with decision support, pharmacy approval of specific drugs before administration, and multiple nursing checks at administration [12]. Additionally, the medication environment of different hospitals (such as physical environment and human environment) has its particularity, which requires nurses to constantly summarise and explore medication experience in practice, strengthen their keen perception of risk factors in the medication environment, and then form the ability to adapt to their medication environment safety. Managers should create a supportive work environment of safe medication in the OR, promote nurses' perception of their medication environment through regular department seminars, and encourage exchanges and learning among medical staff on medication safety experience to get familiar with possible risk factors in the medication environment and countermeasures.

#### 4.2.2 Working years.

It is worth noting that the results of multiple linear regression showed that the longer the working years, the higher the medication safety behaviour of OR nurses. This is consistent with the study by SLOSS et al., which argues that new nurses have a higher incidence of medication errors and are more likely to have unsafe or irregular behaviour in medication management [33]. In addition, relevant studies have shown that nurses who have worked for 5–10 years may be more likely to have unsafe drug use behaviour than nurses who have worked for 15–20 years [25] and newly graduated (less than 1 year of service) may have higher levels of medication insecurity than nurses with 1–5 years of service [26]. Drug management in the OR is difficult, and new nurses are not skilled in drug operation

procedures, which may lead to blind implementation of medical orders. In addition, it may be that nurses with longer working years have more drug administration experience, and they are less susceptible to unsafe medication behaviours. In contrast, younger nurses are relatively less safety conscious and have drug administration experience. Therefore, nursing managers should focus on the drug safety ability of new nurses and train them in the knowledge and operation to make up for their lack of professional knowledge.

### 4.2.3 Highest level of education.

Our studies showed that education is significantly positively correlated with medication-safe behaviour, consistent with HODKINSON et al. [8]. With the development of nursing education, nurses in tertiary hospitals gradually transition to higher education, the knowledge system of nursing education is constantly improved, and the training program of theoretical knowledge and clinical practice of nursing postgraduates goes hand in hand makes the medication safety ability of nursing undergraduates and postgraduates constantly improve. Therefore, young nurses should be encouraged to continue their education, and hospital managers should consider providing opportunities for continuing education based on merit.

### 4.2.4 system and supervision.

*System and supervision* are important predictors of OR nurses' safe medication behaviour. Ensuring strict implementation of a safe medication management system in OR is the core of ensuring safe medication for nurses. Previous studies have shown [26] that nurses have a lower score of unsafe medication behaviour when the department has a sound system and effective supervision. Boytim et al. [9] pointed out that haste, tension and progressive pressure were the high-frequency human factors leading to medication errors by nurses in the operating room. It can be seen that improving the medication safety behaviour of OR nurses is not only the behaviour of individual nurses but also should be included in the continuous improvement of the management system. Therefore, it is necessary to establish a perfect medication management system, such as the execution system of doctor's orders, the drug administration process, and the patient's drug safety monitoring system. Departments should also be called upon to regularly check the validity period and quality of drugs and reasonably arrange the level of nurses in each shift [34]. Additionally, Monitoring and reporting adverse drug events can ensure drug safety and improve medical quality, reflecting the degree of attention paid to drug safety in OR [35]. Relevant surveys showed that more than half of medication errors in the OR are not fully reported [36], and accurate identification and active reporting of medication errors by nurses are crucial to learning from mistakes and reducing similar errors. Therefore, an adverse drug event reporting system should be established to standardise the reporting process and disposal of adverse drug events, and surgeons, anesthesiologists, OR nurses, and other personnel should be encouraged to actively participate in Monitoring and reporting medication errors. According to a survey, clinical decision support algorithms can prevent 95% of self-reported medication errors in OR [37]. Information technology should be used to establish an electronic drug management system to realise the traceability of drug information and monitor and manage the medication process, which will identify and correct potential medication errors promptly.

### 4.2.5 Professional title.

However, the higher the professional title, the worse the medication safety behaviour of the OR nurses. This result was similar to previous studies [25,26]. Although nurses with high professional titles have rich experience and are familiar with the complex clinical environment, they can still exhibit unsafe behaviours due to inertia thinking, set psychology, and temporary negligence. Moreover, nurses with high professional titles are mostly the department's backbone and have multiple roles, making them more susceptible to external interference. In the future, more large-sample studies should be conducted on the group of operating room nurses to determine the impact of professional titles on their drug safety behaviours, and targeted training and education should be conducted for nurses with different professional titles.

### 4.2.6 The last time medication safety training.

This study showed that the closer the training time is, the better the medication safety behaviour of the operating room nurses is. Some studies have found a strong correlation between healthcare workers who attend safety training seminars and increased rates of medication error reporting [38]. Time significantly impacts learning effectiveness, with memory degrading over time, completely forgotten when a threshold is

reached, and learning effectiveness declining over time [39]. With the development of modern medical technology, drugs are updated rapidly, and the types are increasing year by year, which requires OR nurses to constantly understand the latest drug situation, master the principle, route, dosage, toxic and side effects of medication, to reduce the risk caused by adverse drugs. OR managers should regularly invite the heads of departments, anesthesiologists and pharmacists of the hospital to conduct training and lectures on medication safety knowledge so that OR nurses can improve their understanding of drug knowledge and new developments through drug training to reduce risks and ensure the safety of surgical patients.

## 4.1 Limitations

However, the study has some limitations. The nurses' medication environment perception scale lacks the characteristics of OR specialities, and self-designed questionnaires are used to investigate the medication safety behaviour of OR nurses so that the results may be subjective. In the future, professional questionnaires or scientific objective evaluation tools with good reliability and validity and features of operating room specialities should be designed to further evaluate their medication safety behaviour. In addition, the hospitals in this study are all tertiary hospitals, and due to limited human and material resources, the sample size is small and comes from a specific region, which may limit the generality of the findings to other environments or populations. Future studies should conduct a multi-centre investigation to comprehensively explore the factors influencing the medication safety behaviour of OR nurses with different demographic characteristics.

## 5 Conclusion

In summary, the medication safety behaviour of nurses in the OR is at a low level. Medication environment, working years, the highest level of education, professional title, and the last time medication safety training influence medication safety behaviour. In the future, a supportive medication environment in OR should be created based on environmental factors such as department culture and system, relevant personnel and hardware facilities. The medication safety behaviour of OR nurses should be improved in various aspects, such as improving the drug safety management system, strengthening training and assessment, and establishing a safety reporting mechanism to reduce MAEs.

## Author contributions

**Conceptualization:** Xiuwen Chen, Xueyi Wei.

**Data curation:** Xiuwen Chen, Xueyi Wei, Liqing Yue, Duo Wu.

**Funding acquisition:** Xiuwen Chen.

**Investigation:** Xiuwen Chen, Xueyi Wei.

**Methodology:** Xiuwen Chen, Xueyi Wei, Duo Wu.

**Resources:** Xueyi Wei, Liqing Yue.

**Software:** Xueyi Wei.

**Supervision:** Jiqun He.

**Writing – original draft:** Xueyi Wei.

**Writing – review & editing:** Xiuwen Chen, Liqing Yue, Jiqun He.

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
