## [Decision Letter · Decision Letter 0]

17 Dec 2024

PONE-D-24-50882Status and Related Factors of Medication Safety Behavior of Nurses in Operating Room: A Cross-Sectional Survey in ChinaPLOS ONE

Dear Dr. Chen,

Thank you for submitting your manuscript to PLOS ONE. After careful consideration, we feel that it has merit but does not fully meet PLOS ONE’s publication criteria as it currently stands. Therefore, we invite you to submit a revised version of the manuscript that addresses the points raised during the review process.

Please submit your revised manuscript by Jan 31 2025 11:59PM. If you will need more time than this to complete your revisions, please reply to this message or contact the journal office at plosone@plos.org . Please include the following items when submitting your revised manuscript:

We look forward to receiving your revised manuscript.

Kind regards,

Othman A. Alfuqaha, Ph.D.

Academic Editor

PLOS ONE

Journal Requirements:

3. Please ensure that you have specified a) Did participants provide their written or verbal informed consent to participate in this study?

b) If consent was verbal, please explain i) why written consent was not obtained, ii) how you documented participant consent, and iii) whether the ethics committees/IRB approved this consent procedure."

- In consent please state in Ethics Method section and manuscript if it is written or verbal. If consent was verbal, please explain a) why written consent was not obtained, b) how you documented participant consent, and c) whether the ethics committees/IRB approved this consent procedure.

[This work was supported by the 2024 Annual Graduate Student Independent Exploration and Innovation Project of Central South University (2024ZZTS0548)].

5. In the online submission form, you indicated that The data that support the findings of this study are available on request from the corresponding].

6. PLOS requires an ORCID iD for the corresponding author in Editorial Manager on papers submitted after December 6th, 2016. Please ensure that you have an ORCID iD and that it is validated in Editorial Manager. To do this, go to ‘Update my Information’ (in the upper left-hand corner of the main menu), and click on the Fetch/Validate link next to the ORCID field. This will take you to the ORCID site and allow you to create a new iD or authenticate a pre-existing iD in Editorial Manager.

Additional Editor Comments:

Dear Dr. Chen,

Thank you for submitting your manuscript titled Status and Related Factors of Medication Safety Behavior of Nurses in Operating Room: A Cross-Sectional Survey in China to PLOS ONE. The reviewers have provided valuable feedback, and after careful consideration, I have decided that the manuscript requires major revisions before it can be considered further for publication.

Below, I have outlined the key areas that must be addressed:

References and Content Clarification:

- Conduct a thorough revision of the English language to address grammatical and stylistic errors. Consider professional editing services.

- Ensure the study's objectives are clearly addressed in the abstract.

-Replace Reference 9 (Cooper et al., 1984) as it is outdated and pertains to anesthesiologists rather than operating room nurses. Ensure all references are current and relevant to the context of OR nurses in China.

Address the findings by Song et al. (2022) regarding the most common medical errors in operating room settings (e.g., surgical instruments, sterilization) and clarify the relevance of medication errors to your study.

Unreferenced Sections (Lines 67-76):

- Add appropriate references to substantiate claims, particularly those involving medication safety practices.

Clarify whether administration of anesthetic drugs is performed by nurses in the study setting.

- Sample Size and Variables: Clearly justify the sample size and explain how it is sufficient for the study's 11 variables.

- Provide details on the sampling technique used.

- Ethical Considerations: Clarify whether informed consent was obtained from all study participants.

- Include confidence intervals and p-values for all statistically significant results.

- Expand the discussion to provide deeper insights into your findings and align them with relevant current literature.

- Practical Significance: Reassess the practical significance of your findings given the sample size limitations. Discuss any implications for real-world applications.

- Review the manuscript carefully to ensure adherence to PLOS ONE's formatting and writing guidelines.

Best of luck in your revision.

Reviewers' comments:

Reviewer's Responses to Questions

**Comments to the Author**

1. Is the manuscript technically sound, and do the data support the conclusions?

Reviewer #1: No

Reviewer #2: Partly

2. Has the statistical analysis been performed appropriately and rigorously? 

Reviewer #1: No

Reviewer #2: Yes

3. Have the authors made all data underlying the findings in their manuscript fully available?

Reviewer #1: Yes

Reviewer #2: Yes

4. Is the manuscript presented in an intelligible fashion and written in standard English?

Reviewer #1: No

Reviewer #2: Yes

5. Review Comments to the Author

Reviewer #1: Review Comments

Line 56-59: The content mentioned in Reference 9 states that anesthesiologists are responsible for single use of drugs in the operating room, without a double-checking step. The following points need clarification:

The content in Reference 9 (Cooper JB, Newbower RS, Kitz RJ. An analysis of major errors and equipment failures in anesthesia management: considerations for prevention and detection. Anesthesiology 1984; 60: 34–42) is based on a 1984 study. In China, as of 2024, it is rare for anesthesiologists to administer drugs without double-checking; almost all perform double checks.

The study subjects in Reference 9 were anesthesiologists, not operating room nurses, who are the subjects of this article. Recent investigations into medical errors among operating room nurses (Song Q, Tang J, Wei Z, Sun L. Prevalence and associated factors of self-reported medical errors and adverse events among operating room nurses in China. Front Public Health. 2022;10:988134. doi: 10.3389/fpubh.2022.988134) show that the most common medical errors stem from surgical instruments (9.1%), sterilization (9.0%), equipment and consumables (8.9%), and specimen management (7.8%). Thus, medication errors are not among the most common medical errors in the operating room.

Line 67-71: There is no reference to support this section. Considering the previous discussion pertains to anesthesiologists rather than operating room nurses, I suspect the author intended to refer to MAEs caused by anesthesiologists rather than operating room nurses. This leads to a lack of basis for this study and suggests that the author should review this section.

Line 72-76: There is no reference to support this section. It is unclear whether the administration of anesthetic drugs is performed by nurses.

Line 95: The study's 11 variables do not match the research variable count, indicating an insufficient sample size.

Informed Consent: It is unclear whether the study subjects signed informed consent forms.

Correlation Analysis Results: Even if the correlation analysis results are significant, the small sample size means that the observed associations have minimal practical significance in real-world applications.

Reviewer #2: Dear author/s,

The manuscript had the aim to explore the current status and related factors regarding medication safety behavior of operating room nurses in China. The study is important; however, the study has some limitations.

Abstract

The objective of this study was to explore the current status and related factors regarding medication safety behavior of operating room nurses. But, in my opinion, the objectives of this study are not answered well.

Result: It is better to write with confidence intervals and/or p-values of those listed as statistically significant factors.

Keywords: Please write the keywords based on MeSH terms.

I believe that a revision of the English language is necessary because there are a lot of grammatical errors throughout the document.

The authors should consider a review of cited data and follow the PlOS ONE journal authors writing guidelines.

The authors do not write with clear sample size calculation and sampling techniques. Please clarify that.

The result of this study needs revision.

The discussion is poor. The findings could have been better discussed.

6. PLOS authors have the option to publish the peer review history of their article (what does this mean? ). If published, this will include your full peer review and any attached files.

**Do you want your identity to be public for this peer review?** For information about this choice, including consent withdrawal, please see our Privacy Policy .

Reviewer #1: No

Reviewer #2: No

---

## [Author Response · Author response to Decision Letter 0]

15 Jan 2025

Reviewers' comments:

Reviewer's Responses to Questions

Comments to the Author

1. Is the manuscript technically sound, and do the data support the conclusions?

Reviewer #1: No

Answer: Thank you for your valuable feedback.

We have made comprehensive revisions based on the editor's and reviewers' comments to ensure that experiments are conducted rigorously. We have also modified and supplemented the sampling method and sample size calculation as suggested.  We also discussed and explained the limitations of the sample size, and re-evaluated the real-world implications of our findings.

Additionally, our team has rechecked and analyzed the data once more and consulted a statistical expert to verify the accuracy of our data analysis methods.

Reviewer #2: Partly

Answer: Thank you for your review.

We have made comprehensive revisions based on the editor's and reviewers' comments to ensure that experiments are conducted rigorously. We have also modified and supplemented the sampling method and sample size calculation as suggested.  We also discussed and explained the limitations of the sample size, and re-evaluated the real-world implications of our findings.

Additionally, our team has rechecked and analyzed the data once more and consulted a statistical expert to verify the accuracy of our data analysis methods.

2. Has the statistical analysis been performed appropriately and rigorously?

Reviewer #1: No

Answer: Thank you for your valuable feedback.

We have made comprehensive revisions based on the editor's and reviewers' comments to ensure that the statistical analysis has been performed appropriately and rigorously. Additionally, our team has rechecked and analyzed the data once more and consulted a statistical expert to verify the accuracy of our data analysis methods.

Reviewer #2: Yes

Answer: Thank you for your positive feedback.

3. Have the authors made all data underlying the findings in their manuscript fully available?

Reviewer #1: Yes

Answer: Thank you for your positive feedback.

Reviewer #2: Yes

Answer: Thank you for your positive feedback.

4. Is the manuscript presented in an intelligible fashion and written in standard English?

Reviewer #1: No

Answer: Thank you for highlighting this important point.

We have made revisions based on your feedback and sought the assistance of a professional service for language editing to improve clarity and correctness. We appreciate your guidance and welcome any further concerns you may have.

Reviewer #2: Yes

Answer: Thank you for your positive feedback.

5. Review Comments to the Author

Answer: Thank you for your thorough review and comments.

We have carefully considered your feedback and made the necessary revisions to address the issues raised. We sincerely appreciate your attention to detail and guidance in improving the manuscript. If you have any further concerns or additional comments, please do not hesitate to let us know.

Reviewer #1: Review Comments

Line 56-59: The content mentioned in Reference 9 states that anesthesiologists are responsible for single use of drugs in the operating room, without a double-checking step. The following points need clarification:

The content in Reference 9 (Cooper JB, Newbower RS, Kitz RJ. An analysis of major errors and equipment failures in anesthesia management: considerations for prevention and detection. Anesthesiology 1984; 60: 34–42) is based on a 1984 study. In China, as of 2024, it is rare for anesthesiologists to administer drugs without double-checking; almost all perform double checks.

Answer: Thank you for your valuable feedback.

We fully understand your concern regarding Reference 9 (Cooper JB, Newbower RS, Kitz RJ. An analysis of major errors and equipment failures in anesthesia management: considerations for prevention and detection. Anesthesiology 1984; 60: 34–42). After carefully considering your comments, we have decided to delete this reference and the content of the citation from our manuscript.  Additionally, we have rewritten the second paragraph of the introduction(Line 55-72 in the revised manuscript with track changes)

The study subjects in Reference 9 were anesthesiologists, not operating room nurses, who are the subjects of this article. Recent investigations into medical errors among operating room nurses (Song Q, Tang J, Wei Z, Sun L. Prevalence and associated factors of self-reported medical errors and adverse events among operating room nurses in China. Front Public Health. 2022;10:988134. doi: 10.3389/fpubh.2022.988134) show that the most common medical errors stem from surgical instruments (9.1%), sterilization (9.0%), equipment and consumables (8.9%), and specimen management (7.8%). Thus, medication errors are not among the most common medical errors in the operating room.

Answer: Thank you for your careful review.

We appreciate your pointing out that the study subjects in Reference 9 were anesthesiologists, not operating room nurses, who are the focus of our article. After carefully reading Reference 9(Cooper JB, Newbower RS, Kitz RJ. An analysis of major errors and equipment failures in anesthesia management: considerations for prevention and detection. Anesthesiology 1984; 60: 34–42), we decided to delete the reference and the quoted content.

Additionally, we acknowledge the recent investigations into medical errors among operating room nurses, as highlighted by the study by Song et al. (2022), which show that the most common medical errors in the operating room stem from surgical instruments, sterilization, equipment and consumables, and specimen management, rather than medication errors. We have carefully read the article. Although Song et al reported that medication errors are not the most common medical errors of operating room nurses, compared with other departments, there are many high-risk drugs in operating rooms, and management is difficult. Medication errors will bring serious harm to patients, and most medication errors are preventable. Therefore, it is very necessary to explore the problem of medication errors in the operating room. In addition, we have reorganized the logic of the Introduction with reference to your valuable comments, and focused this part(Line 55-73 in the revised manuscript with track changes) on the harm of medication errors in the operating room to highlight the necessity of our research.

Thank you again for your valuable feedback. If you have any further questions or concerns, please feel free to reach out to us.

Line 67-71: There is no reference to support this section. Considering the previous discussion pertains to anesthesiologists rather than operating room nurses, I suspect the author intended to refer to MAEs caused by anesthesiologists rather than operating room nurses. This leads to a lack of basis for this study and suggests that the author should review this section.

Answer: Thank you for your insightful suggestions.

We have read this section carefully and reworked the logic. After discussion we have decided to delete Line 67-71 in order to avoid controversy. We have rewritten the second paragraph of the introduction(Line 55-72 in the revised manuscript with track changes), and focused this part on the harm of medication errors in the operating room. After introducing the harm of medication errors in the operating room, we then described the important role of the operating room nurses in the medication process, so as to introduce the importance of improving the safe medication behavior of the operating room nurses, thus highlighting our research theme.

Line 72-76: There is no reference to support this section. It is unclear whether the administration of anesthetic drugs is performed by nurses.

Answer: Thank you for raising this point.

We understand your concerns about the lack of reference support for this section and the uncertainty regarding the role of nurses in administering anesthetic drugs. Firstly, regarding the lack of references, we acknowledge that we omitted relevant literature in our previous version. 

Line 73-77 in the revised manuscript with track changes: We added reference 8 (Suzuki R, Imai T, Sakai T, Tanabe K, Ohtsu F. Medication Errors in the Operating Room: An Analysis of Contributing Factors and Related Drugs in Case Reports from a Japanese Medication Error Database. J Patient Saf 2022;18(2):e496-e502) and reference 16 (Yin L, Chen S, Yu J. Discussion on medication risk management for nurses in operating room. Journal of Chinese medicine administration 2019;27(20):139-141).

We have also refined this part to make it more clear. Operating room nurses play a crucial role in the drug administration process, mainly including dispensing drugs, checking drugs, and assisting the anesthesiologist to administer drugs. The operating room nurse serves as an assistant to assist the anesthesiologist in administering anesthetic drugs and performing appropriate intraoperative administration as instructed by the doctor.

Line 95: The study's 11 variables do not match the research variable count, indicating an insufficient sample size.

Answer: Thank you for your careful review.

First, we sincerely apologize for the error. To ensure data accuracy, our team carefully double-checked all the variables. Unfortunately, we discovered an error in the variables of the original manuscript. But we have revisioned it at the section 2.1 “Study Design and participants” and it should be 15 independent variables (10 demographic variables, five dimensions of nurses' medication environment perception scale). We have added specifics to the 15 variables.

For the sample size, the recommendations of between 5 and 10 times the number of independent variables and considering that 20% of invalid questionnaires, a minimum of 90 operating nurses was required. We hope this clarifies the situation, and we appreciate your understanding. If you have any further questions or concerns, please feel free to reach out to us.

Informed Consent: It is unclear whether the study subjects signed informed consent forms.

Answer: Thank you for raising this point.

Upon carefully examining your comments, we realize that there may have been some misunderstanding or unclear statement in our description of manuscript.

We described the informed consent of study subjects in the original manuscript. For exampleLine 152-153: The study complied with the Declaration of Helsinki and all participants gave informed consent and their participation was anonymous.

Line91-92: (4) informed consent and voluntary participation in the study.

In addition, our previous descriptions may not have been clear enough, in reponse to your suggestion, we added that “all participants signed informed consent” at the section of 2.5 “Ethical considerations” of line 157 in the revised manuscript with track changes.

Correlation Analysis Results: Even if the correlation analysis results are significant, the small sample size means that the observed associations have minimal practical significance in real-world applications.

Answer: Thank you for your valuable feedback.

Your point is well-taken, and we acknowledge that the small sample size may limit the generality of our findings in real-world applications. The small sample size is a major limitation of our article. Although the correlation analysis results are significant, they still need to be interpreted with caution. We have reassessed the practical significance of our findings given the sample size limitations. In the discussion part, a large number of references are cited as evidence and the sample size problem is further explained, so that readers can carefully consider our research results. We also discuss this in the section Limitations of the revised manuscript. To enhance the practical significance of our research, we plan to expand the sample size in future studies and validate our findings in different populations and settings. This will help to more comprehensively assess the relationships between variables and provide stronger evidence for their effectiveness in real-world applications. Once again, thank you for your valuable comments. We have taken your suggestions into consideration and will strive to improve our research in subsequent studies.

Reviewer #2: Dear author/s,

The manuscript had the aim to explore the current status and related factors regarding medication safety behavior of operating room nurses in China. The study is important; however, the study has some limitations.

Abstract

The objective of this study was to explore the current status and related factors regarding medication safety behavior of operating room nurses. But, in my opinion, the objectives of this study are not answered well.

Result: It is better to write with confidence intervals and/or p-values of those listed as statistically significant factors.

Answer: Thank you for your valuable feedback

We have carefully reviewed the abstract and have made revisions to the aim and results sections in line with your suggestion to ensure the study's objectives are clearly addressed in the abstract.

Additionally, we have added p-values in the result section of abstract.

Keywords: Please write the keywords based on MeSH terms.

Answer: Thank you for your valuable feedback

We have searched the Mesh terms thesaurus and reformulated the keywords to ensure that all the keywords are Mesh terms.

I believe that a revision of the English language is necessary because there are a lot of grammatical errors throughout the document.

Answer: Thank you for highlighting this important point.

We have made revisions based on your feedback and sought the assistance of a professional service for language editing to improve clarity and correctness. We appreciate your guidance and welcome any further concerns you may have.

The authors should consider a review of cited data and follow the PlOS ONE journal authors writing guidelines.

Answer:Thank you for your suggestion.

We have carefully reviewed our cited data to ensure its completeness and accuracy. We can confirm that no retracted papers have been cited in this manuscript. All referenced literature is valid, relevant, and supports our research viewpoints and conclusions.

We have carefully reviewed the PLOS ONE style requirements and ensured that both our manuscript and file names fully comply with the provided guidelines.

The authors do not write with clear sample size calculation and sampling techniques. Please clarify that.

Answer: Thank you for your valuable suggestion.

We understand your concern and appreciate your guidance in improving our manuscript. In response to your suggestions, we have made the necessary additions at the section 2.1 “Study Design and participants” in the revised manuscript with track chang

---

## [Decision Letter · Decision Letter 1]

17 Feb 2025

Status and Related Factors of Medication Safety Behavior of Nurses in Operating Room: A Cross-Sectional Survey in China

PONE-D-24-50882R1

Dear Dr.

<table border="0" cellpadding="0" cellspacing="0" class="datatable3" style="border-collapse: collapse; width: 678px; line-height: 14px; color: rgb(0, 0, 51); font-family: verdana, geneva, arial, helvetica, sans-serif; font-size: 11.2px;"> <tbody> <tr style="background-color: rgb(244, 244, 244);"> <td style="padding: 3px; border: 1px solid rgb(255, 255, 255);">Liqing Yue</td> </tr> <tr style="background-color: rgb(244, 244, 244);"> <td style="padding: 3px; border: 1px solid rgb(255, 255, 255); width: 196.094px;"> </td> </tr> </tbody></table>

We’re pleased to inform you that your manuscript has been judged scientifically suitable for publication and will be formally accepted for publication once it meets all outstanding technical requirements.

Kind regards,

Othman A. Alfuqaha, Ph.D.

Academic Editor

PLOS ONE

Additional Editor Comments (optional):

Dear authors,

I am pleased to inform you that your manuscript, "Status and Related Factors of Medication Safety Behavior of Nurses in Operating Room: A Cross-Sectional Survey in China," has been accepted for publication.

Congratulations on this achievement! Your work provides valuable insights into medication safety behavior among nurses, and I am confident it will contribute significantly to the field.

Wishing you continued success in your research endeavors.

Best regards,

Dr. Alfuqaha

Reviewers' comments:

Reviewer's Responses to Questions

**Comments to the Author**

1. If the authors have adequately addressed your comments raised in a previous round of review and you feel that this manuscript is now acceptable for publication, you may indicate that here to bypass the “Comments to the Author” section, enter your conflict of interest statement in the “Confidential to Editor” section, and submit your "Accept" recommendation.

Reviewer #2: (No Response)

Reviewer #3: All comments have been addressed

2. Is the manuscript technically sound, and do the data support the conclusions?

Reviewer #2: (No Response)

Reviewer #3: Yes

3. Has the statistical analysis been performed appropriately and rigorously? 

Reviewer #2: (No Response)

Reviewer #3: Yes

4. Have the authors made all data underlying the findings in their manuscript fully available?

Reviewer #2: (No Response)

Reviewer #3: Yes

5. Is the manuscript presented in an intelligible fashion and written in standard English?

Reviewer #2: (No Response)

Reviewer #3: Yes

6. Review Comments to the Author

Reviewer #2: Dear authors, Thank you so much for your invaluable work. My comments were fully addressed and will be accepted for publication.

Reviewer #3: After careful review, I am pleased to inform you that all concerns have been adequately addressed. The improvements in statistical justifications, discussion clarity, results presentation, and language refinement have strengthened the manuscript.

I am happy to recommend its acceptance in the current form. Congratulations on your work!

7. PLOS authors have the option to publish the peer review history of their article (what does this mean? ). If published, this will include your full peer review and any attached files.

**Do you want your identity to be public for this peer review?** For information about this choice, including consent withdrawal, please see our Privacy Policy .

Reviewer #2: No

Reviewer #3: **Yes: ** Mohd Ismail Ibrahim

---

## [Editor Report · Acceptance letter]

PONE-D-24-50882R1

PLOS ONE

Dear Dr. Yue,

I'm pleased to inform you that your manuscript has been deemed suitable for publication in PLOS ONE. Congratulations! Your manuscript is now being handed over to our production team.

Kind regards,

on behalf of

Dr. Othman A. Alfuqaha

Academic Editor

PLOS ONE